# Stacking Rocks to Transport Water: Folk Aqueduct Bridges of Mallorca and Spanish Colonial California

**William E. Doolittle**

Department of Geography and the Environment, The University of Texas at Austin, Austin, TX 78712, USA; dolitl@austin.utexas.edu

**Abstract:** The landscape of Mallorca, Spain is characterized by a number of features constructed of rock. Windmills and walls are ubiquitous and visually striking. Equally widespread, but not as conspicuous, are other features associated with canal irrigation. One such feature that is understudied and therefore underappreciated is that of folk aqueduct bridges. This study investigates these features because they were critical in sustaining agriculture on the island for centuries, because they deserve recognition in order to be preserved as part of the island's cultural and historical heritage, and because of their being antecedents or prototypes of similar structures built in Spanish colonial California. Two field seasons were devoted to locating and studying folk aqueduct bridges. Systematic windshield surveys were undertaken to identify such features. Once located, each folk aqueduct bridge was subjected to detailed description and analysis of size, shape, function, materials, and method of construction. Folk aqueduct bridges of Mallorca were built of shaped and unshaped stone, with channels made of ceramic tiles or ashlar tablets. Many of the rock walls once served as folk aqueduct bridges. Several California missionaries in the 18th century came from Mallorca, and the folk aqueduct bridges they built are based on those of their homeland.

**Keywords:** aqueduct bridges; folk construction; landscapes; irrigation; Mallorca; Spanish California

## 1. Introduction

The term "aqueduct" typically animates either of two images, one specific, and one general. The specific image is that of large, arched, masonry structures typically associated with the ancient Romans. The more general image is that of a long water conveyance system that begins at a source, such as a spring, and terminates on fields of crops or in cities. In such systems, canals or pipes extend along the surface, through tunnels, under hills, and over low-lying areas by means of bridge-like structures. The specific bridge-like structures are distinguished from the generalized systems by use of the term "aqueduct bridges" [1] (pp. 2010–2012). The construction of large aqueduct bridges typically involves a high degree of planning, architectural expertise, and engineering knowledge. However, aqueduct bridges need not be large, arched, masonry, nor Roman. Some consist simply of earth and rock piled into the gaps between high points in order to maintain a constant gradient for the flow of water from one place to another by means of gravity [2]. Such features were built with minimal planning, a modicum of engineering knowledge, and no architectural expertise. They were built by individual farmers or small groups of laborers in an age before modern influences [3], and hence fit the definition of being "folk" structures [4] (pp. 33–38). As such, these small and seemingly less sophisticated aqueduct bridges often go unrecognized and, therefore, unappreciated. One place where folk aqueduct bridges are hidden in plain sight is on the Spanish island of Mallorca in the western Mediterranean Sea (Figure 1).

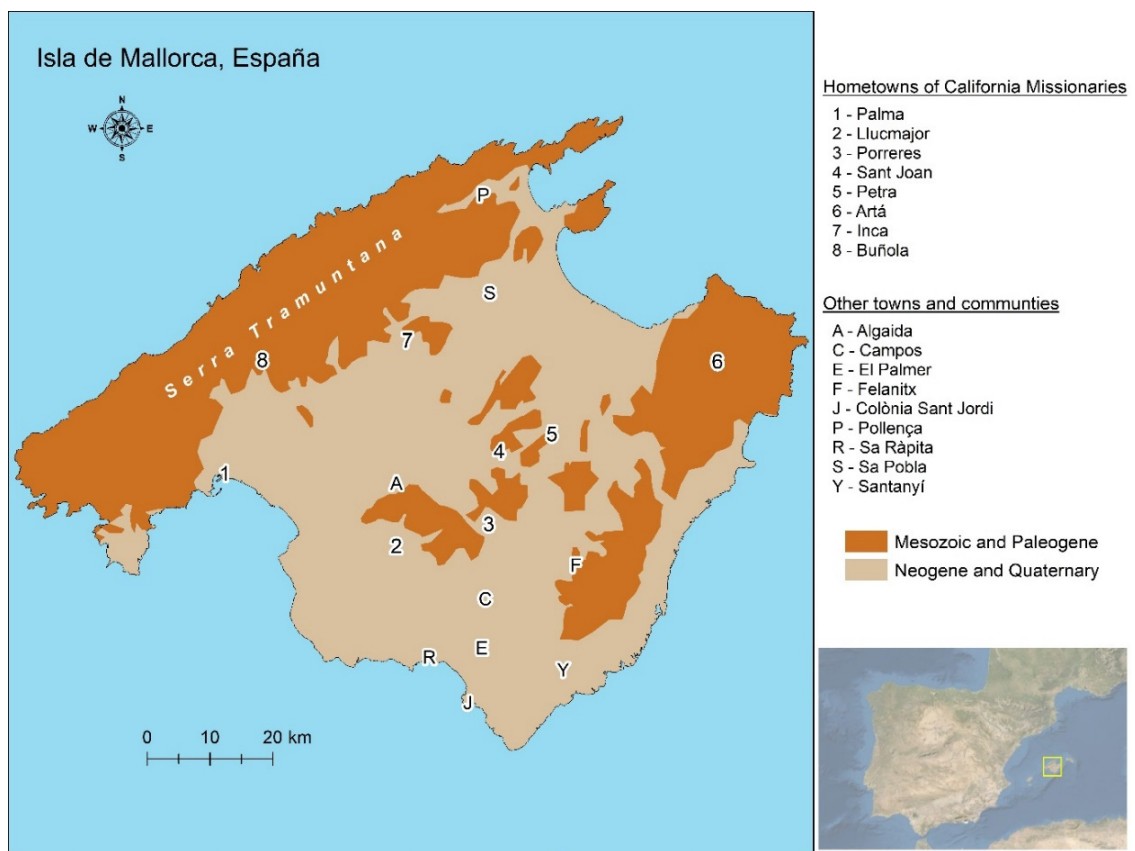

**Figure 1.** Map of Mallorca showing general geology, towns mentioned in the text, and hometowns of missionaries who served in Spanish colonial California. Imagery data: F. Ochoa, 2020.

Numerous studies have focused on Mallorcan irrigation, particularly the lifting of groundwater, and systems delivering water to fields. The former has attracted attention to the devices themselves, their technology, and their construction [5]. The latter has dealt with the complexities of water management and the social implications [6]. Few studies exist on various elements of hydraulic systems. Indeed, Helena Kirchner and Carmen Navarro, two prominent authors on historic Mallorcan water control, stated the case most eloquently: "*El vaciado de referencias a elementos que constituyen los sistemas hidráulicos es, por sí sólo, clarament insuficiente,*" or "The paucity of references to elements that constitute the hydraulic systems is, by itself, clearly insufficient" [7] (p. 98). This paper focuses on one such element: The construction of the long-overlooked folk aqueduct bridges. It does so for four specific reasons pertaining to sustainability. First, the cultural landscape of which they are a part has been classified as having "very high environmental value" and is "very threatened" because of salinization due to the over-pumping of groundwater resulting from post-1960 tourism and development [8] (p. 282). Second, a call has been made by the European Landscape Convention to protect the rock walls of Mallorca, some of which comprise folk aqueduct bridges [9]. Third, their presence is evidence of successful, long-term agriculture. Fourth, their importance for understanding technology transfers elsewhere, particularly to Spanish colonial California.

Aqueduct bridges were constructed throughout what was the Viceroyality of New Spain (now México and the southwestern United States) from the 16th to the 18th century [10]. People involved in their construction came mainly from southern and central Spain, with very few coming from eastern Spain until the 19th century [11]. The single exception to this migration pattern involved clergy, with no fewer than 16 of the missionaries in California coming from Mallorca in the late 18th century [12]. That two of the California missions had folk aqueduct bridges raises the question of their possible Mallorcan origins. The goal of this paper is to better understand and draw attention

to the folk aqueduct bridges of Mallorca, and to explore their being prototypes for those in Spanish colonial California.

## 2. Materials and Methods

This study is part of a long-term project examining the construction of aqueduct bridges in colonial New Spain [13,14] that grew out of an assessment of pre-European water control in that region [15]. Field work constitutes a major component of this project, including early trips to inspect aqueduct bridges in California. A field study of water control features on Mallorca was deemed essential in order to verify or refute technological connections between there and California. Two field seasons were devoted to this endeavor, one in 1997 and one in 2015. Both began with systematic windshield surveys [16] that involved driving along every road in the southern portion of the island and numerous roads elsewhere. Most roads were traversed in both directions and at different times of the day. These surveys were designed to search for, and identify, water control features, specifically folk aqueduct bridges that may be representative of antecedents to those built in colonial New Spain, and particularly those in California. In addition to water control features, related structures, such as walls, rural houses, and outbuildings, were also studied.

Once located, individual features were subjected to detailed investigation. Notes about the materials used in construction, building techniques, and function were recorded. Photographs were taken in 1997 and digital images were recorded in 2015. Unfortunately, the ages of features could not be determined. It is well known that Romans [8] and North Africans [6,7] occupied and built water control structures on Mallorca, and, therefore, the origins of folk aqueduct bridges might well be attributed to them. An investigation into their origins is, however, beyond the scope of this study.

## 3. Results

### 3.1. The Mallorcan Landscape

The cultural landscape of rural Mallorca has been characterized as "a forest of windmills" [17] (p. 281) (Figure 2). In addition to windmills (*molí de vent*) [18], rock walls or stone fences [19], a unique construction technique [20], and olive and wheat cultivation [21,22] contribute to the island's character. Although separate from each other, there exists a common denominator between these four types of features—water [23]. The physical landscape of the island consists of two distinct physiographic regions [8,17]. The wetter northwestern quarter of the island, the Serra de Tramuntana, is mountainous and comprised primarily of hard, folded, impervious Mesozoic and Paleogene limestones [24]. The drier central and southern plains (*Es Pla*) are underlaid by similar limestones in the far east and in scattered locales in the center of the island, but everywhere else are comprised of soft, permeable, and horizontal Miocene limestones and sandstones, as well as Plio-Quaternary deposits [25,26] (Figure 1). The porosity of these rocks has resulted in the area being described as "like a sponge" [27] (p. 149). Most of the 550 mm (22 in) of annual average precipitation percolates quickly through the rock, creating abundant groundwater reserves that are then used to irrigate crops [28]. These eastern plains are the setting for this study.

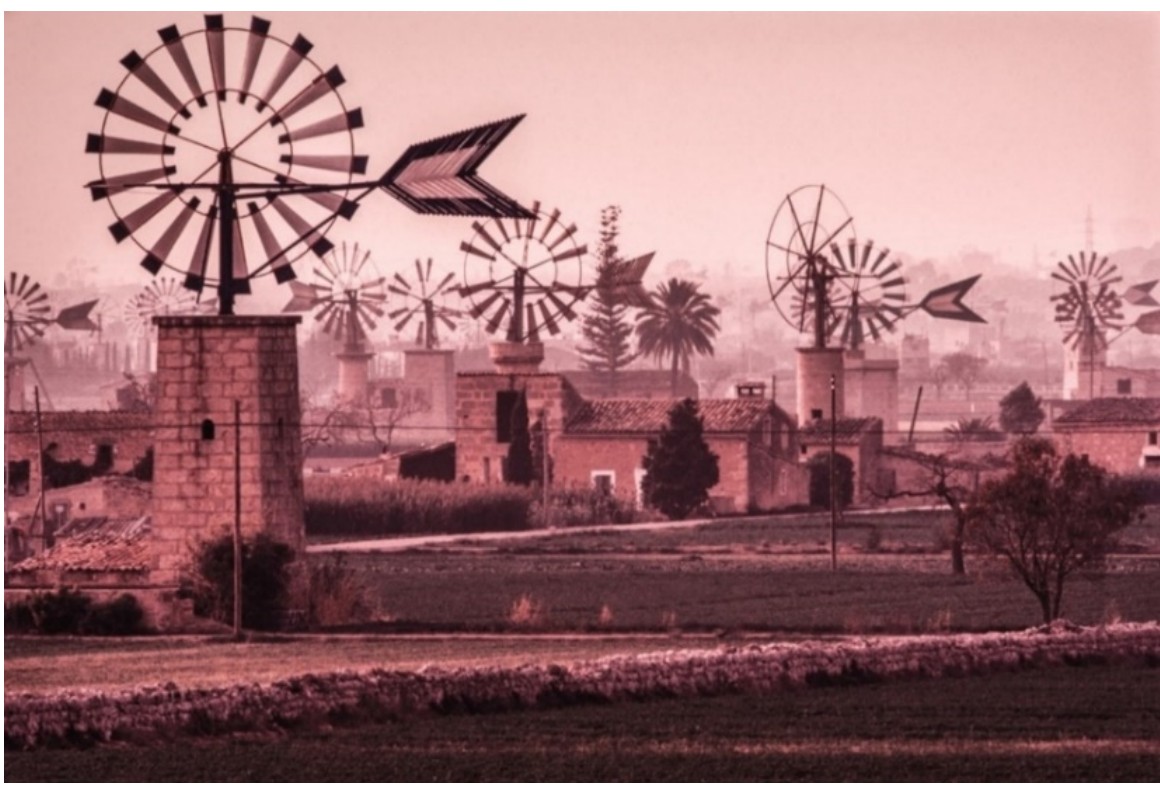

**Figure 2.** A forest of windmills. Imagery data: Reed Kaestner, Getty Images.

### 3.2. Building Materials and Techniques

Although windmills are visually the most striking features on the Mallorcan landscape, houses, smaller structures, and walls are ubiquitous. They are so commonplace, in fact, that their presence alone suggests that both the materials and techniques used in their construction merit investigation in order to gain a fuller appreciation of their relevance for understanding folk aqueduct bridges, their sustainability, and their historic implications.

Rural structures on Mallorca are "born among the people" (hence 'folk' in the title) and, therefore, demonstrate "a practical rather than aesthetic approach to building" [27] (p. 82). They are made of two types of materials—rock rubble (*reble* in Catalan) [27], and ashlar limestone (*calcària*) and sandstone (*marè*) [20]. As throughout much of the world, unshaped rocks of various sizes—rubble—are stacked in order to construct all types of walls. Most rubble walls on Mallorca serve as field boundaries (*fites*) or terrace risers (*marges*). They are built of rocks available in situ, often from field clearance [20,27]. These walls are of dry stone construction, meaning they are built without mortar [19] (Figure 3). The walls of buildings and other features that need support in order to retain their integrity are built with both rock and mortar. Such construction is known as either *mampostería de piedra natural* or simply *mampostería* [29] or *calicanto* [30], literally meaning rock rubble and mortar (Figure 4). Ashlar limestone and sandstone are the second type of material used in wall construction on Mallorca. Quarries exist in a few places on the plain, but the largest is in the far southeast near the town of Santanyí [25,27]. These rocks are so soft that local stonemasons (*picapedreres*) [27] can cut them into ashlar slabs or tablets using handsaws [20,27]. Such tools were used in the past by individual builders (Figure 5), but, today, workshops use large circular saws and sell their product to builders (Figure 6). In addition to quarrying stone for construction, many ashlar tablets appear to have been reused perhaps several times over. This is most evident by the chipped and worn edges of some quoins where they meet the *mampostería* walls on many structures (Figure 4) and on many free-standing walls (Figure 7).

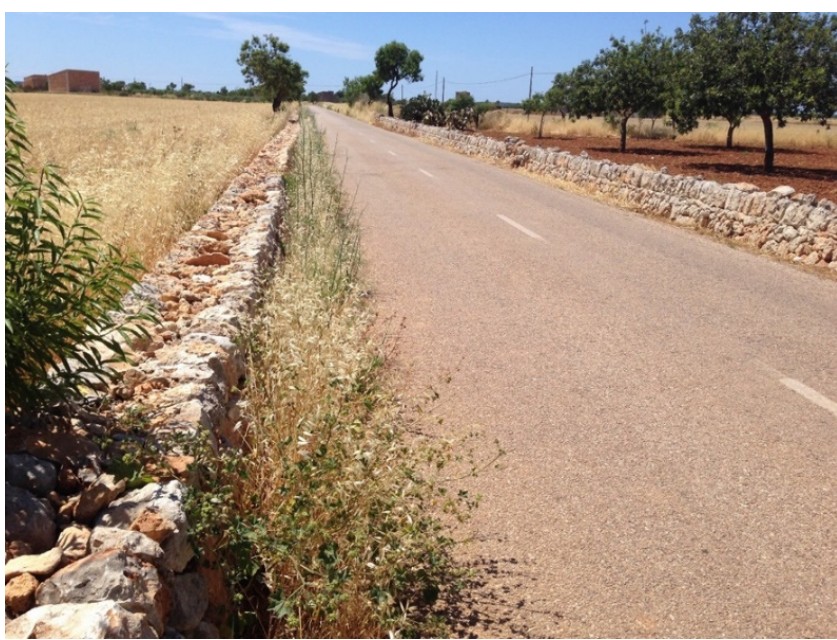

**Figure 3.** Typical dry stone or rock rubble walls on Mallorca. Imagery data: W. Doolittle, 2015.

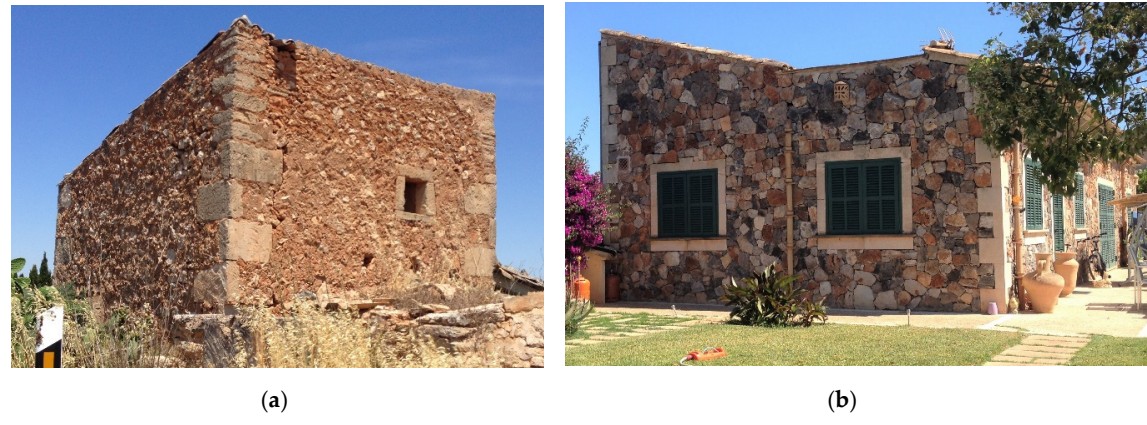

(**a**)  (**b**)

**Figure 4.** Two *mampostería* structures on Mallorca: (**a**) An old outbuilding; (**b**) a new house. Both structures have ashlar tablet quoins (*comús*), a trait nearly unique to Mallorca. Imagery data: W. Doolittle, 2015.

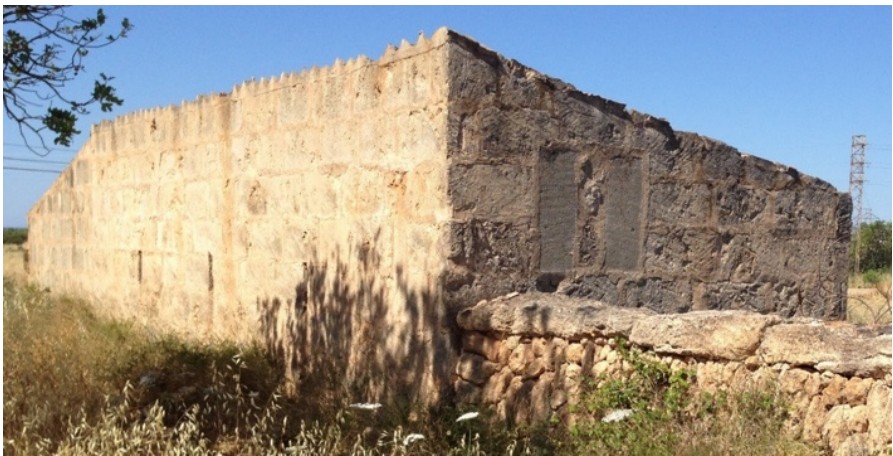

**Figure 5.** An old outbuilding made of (reused?) ashlar tablets cut with a handsaw. Imagery data: W. Doolittle, 2015.

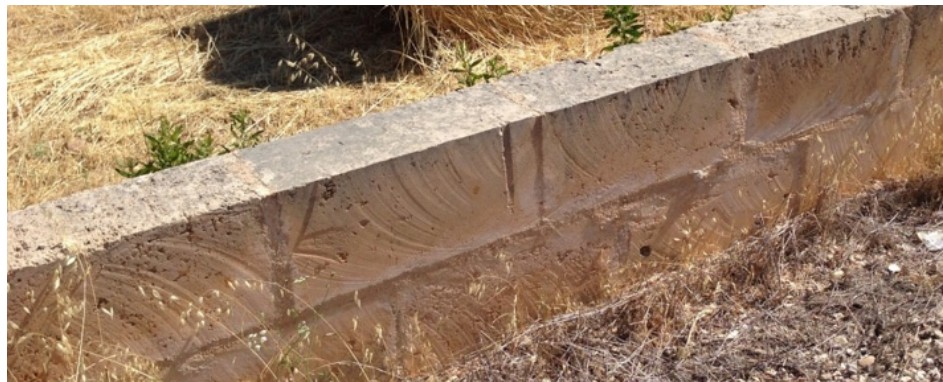

**Figure 6.** Circular saw marks on fresh, new ashlar blocks. Imagery data: W. Doolittle, 2015.

Although ashlar rock is used in many places around the world, the unique characteristic of its use on Mallorca is the way it is cut and placed during construction. Whereas rock is usually cut into rectangular blocks, on Mallorca, it is traditionally cut into tablets that vary in size, but average approximately 75 × 49 × 15 cm. These are then stacked edge to edge or vertically [17] (Figure 7)

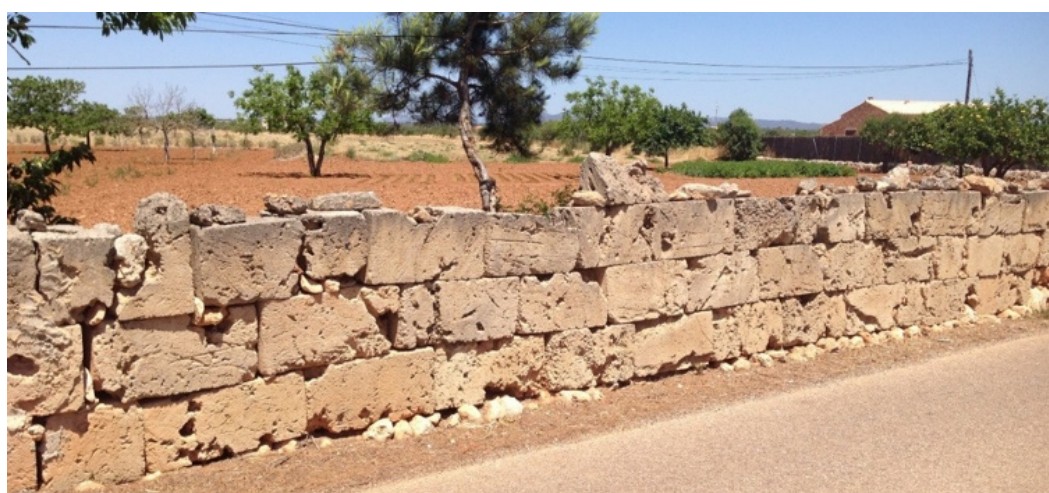

**Figure 7.** An old wall made of reused ashlar tablets of an unknown age. Imagery data: W. Doolittle, 2015.

*3.3. Windmills and Water*

Many of Mallorca's windmills once served as facilities to grind grain into meal and flower, but most once pumped irrigation water to otherwise dry fields [5]. Nearly all of the windmills that existed in 1997 were derelict and in various states of disrepair. One, however, was still functioning (Figure 8). A detailed investigation of that windmill and associated components of its irrigation system, including its folk aqueduct bridge, was undertaken, as was an interview with its owner/operator, who happened to be onsite.

This windmill is located 2 km northeast of the center of Algaida, off of route MA-3131 on the way to Sant Joan (Figure 1). It is easily found on Google Earth at 39°34'12"N 2°54'40"E. It is a square structure made of ashlar tablets (*torre de carreduada de mares i la planta quadrada*) built to lift ground water (*molí elevador d'aigua* or *molí de pujar aigua*) to the surface [5,18]. The turbine was in fine shape and turning, but was not pumping water at the moment, as the square reservoir or storage tank (*safareig*) was already full (Figure 9). Were fields being irrigated at the time, water would have flowed from the tank into an open canal (*sèquia*) and over a small aqueduct bridge comprised of a series of ashlar tablets [27] (Figure 10).

The short drive from the paved road to the windmill was as eye-opening as the windmill itself, and one that required walking and re-walking for closer inspection. The gradient of this 125 m unpaved road was slightly downslope to the windmill (Figure 11). The terminus of the irrigation canal that paralleled the road along its west side was near the intersection of the paved and unpaved roads, and at ground level. Viewed from the windmill, upslope to the paved road, the irrigation water flowed through a canal that was first supported by ashlar tablets (Figure 10) and then ended at field level (Figure 12). The gradient of this canal was downslope, as required for gravity flow, and opposite the gradient of the ground surface.

The owner/operator of this farm and its irrigation system was not the person who built or oversaw its construction. He did, however, say that he has conducted regular maintenance and rebuilding, including cutting and replacing ashlar tablets. His comments affirmed that this is indeed a "folk" system [4].

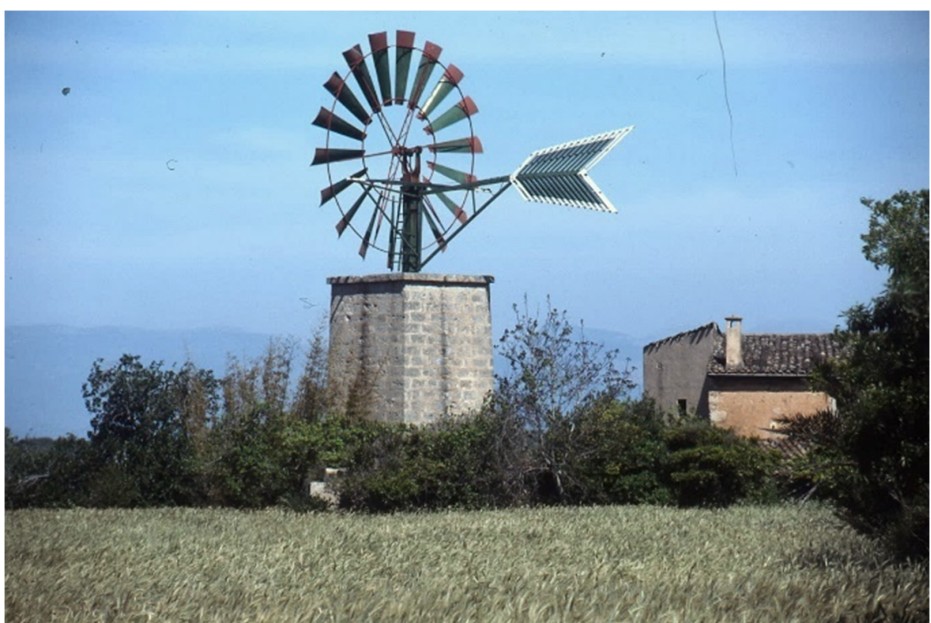

**Figure 8.** The only working water-lifting windmill identified on Mallorca during the first field season. Imagery data: W. Doolittle, 1997.

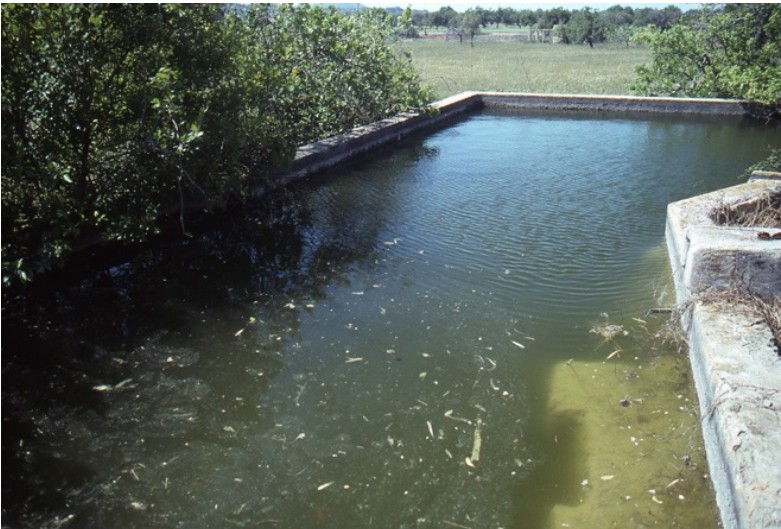

**Figure 9.** The tank in which water pumped from the ground is stored before being released onto the small aqueduct bridge and then onto fields of crops. Imagery data: W. Doolittle, 1997.

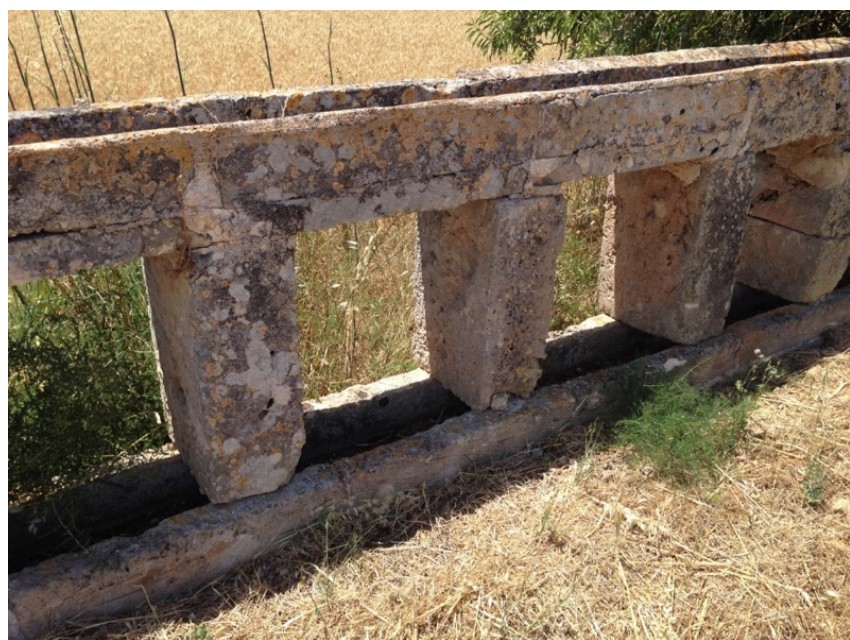

**Figure 10.** A view of the ashlar tablets constituting part of the small aqueduct bridge. Imagery data: W. Doolittle, 2015.

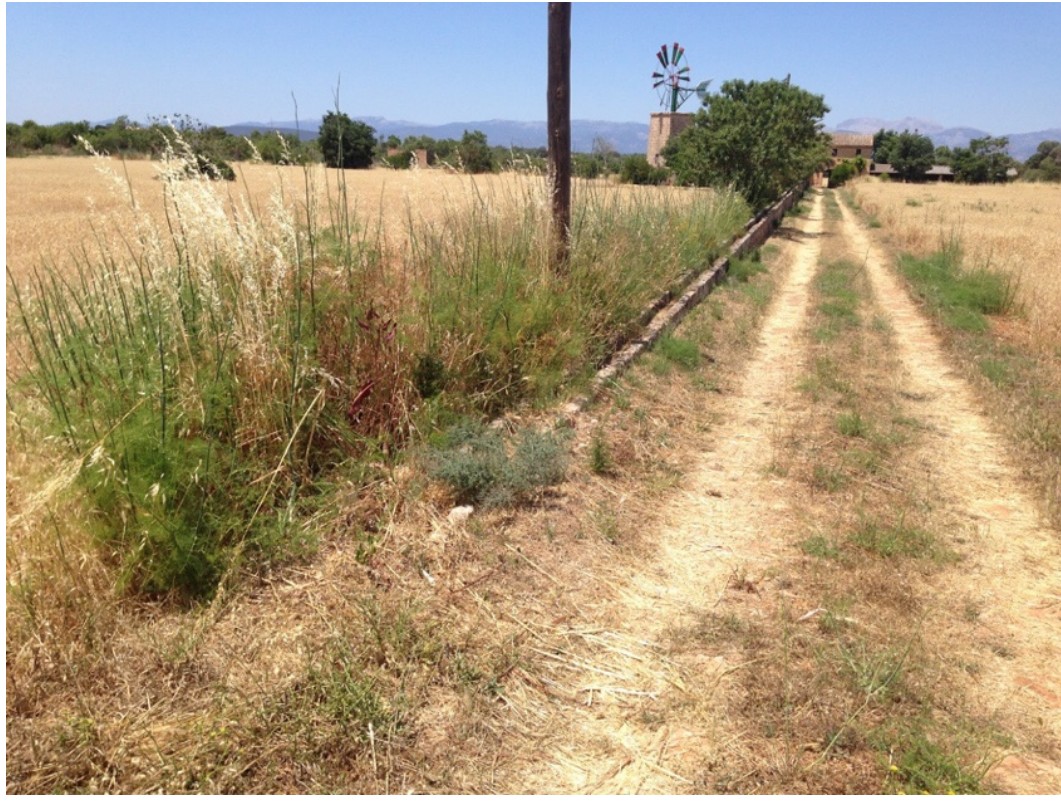

**Figure 11.** A view from the paved road to the windmill in 2015. This system is clearly no longer in use, as evidenced by the missing blades on the turbine (compare to Figure 8) and the prolific growth of tall grasses on the field side of the canal. Imagery data: W. Doolittle, 2015.

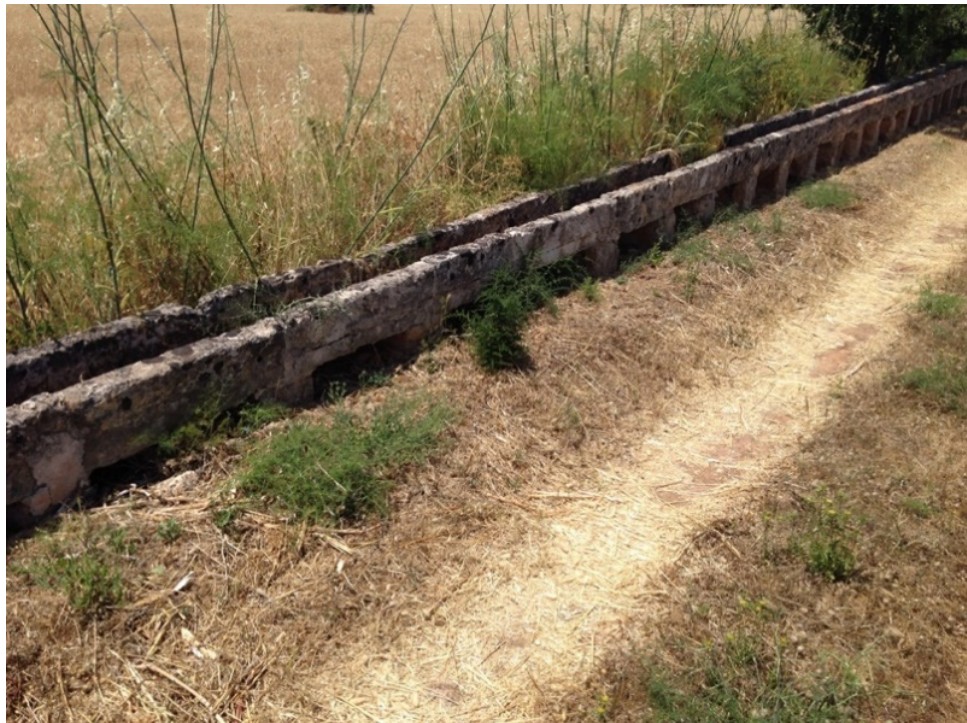

**Figure 12.** A view of the point where the aqueduct bridge ends at ground level. Imagery data: W. Doolittle, 2015.

Folk as the aqueduct bridge near Algaida might be, it was not immune to modernization. The rectangular canal across its top was made from ashlar tablets and required regular maintenance to prevent or at least minimize leakage. A simple and modern solution to this problem was to convert the canal into a place in which a PVC pipe could be laid (Figure 13a). By itself, this is not surprising.

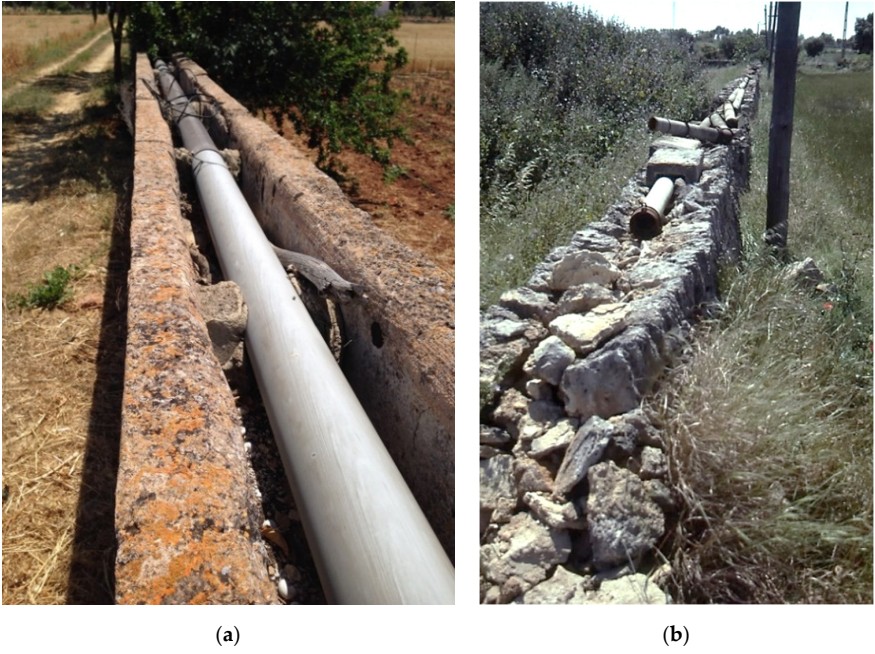

(**a**)                                                (**b**)

**Figure 13.** Old structures, new technology: (**a**) PVC pipe in folk aqueduct bridge near Algaida; (**b**) sections of interlocking metal irrigation pipes placed on a concave-top rock wall near Felanitx (Figure 1). Imagery data: W. Doolittle, (**a**) 2015, (**b**) 1997.

Curious, however, is the widespread practice of placing irrigation pipes atop rock walls or stone fences that run downslope across the island (Figures 13b and 14). If placing pipe in a known canal reflects an old structure adapted to new technology, does pipe placed on old walls indicate that walls once conveyed water across their tops? The shape of many walls suggests so. As is evident in Figures 3, 13b and 14, a large number of walls across Mallorca have concave tops. If these walls had conveyed water across their tops, an impervious canal, perhaps one made of plaster (*guix*), a natural cement (*ciment mallorquí*), or some other material, would have been required [27]. Many of the concave-top walls on Mallorca have been refitted recently using concrete coping (Figure 15).

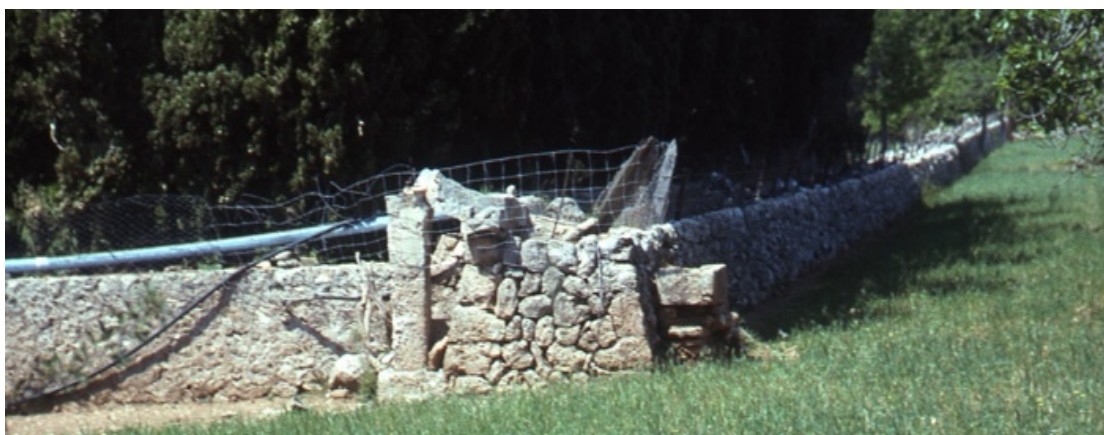

**Figure 14.** A concave-top rock wall supporting an irrigation pipe along route MA-2200 from Sa Pobla to Pollença (Figure 1). Imagery data: W. Doolittle, 1997.

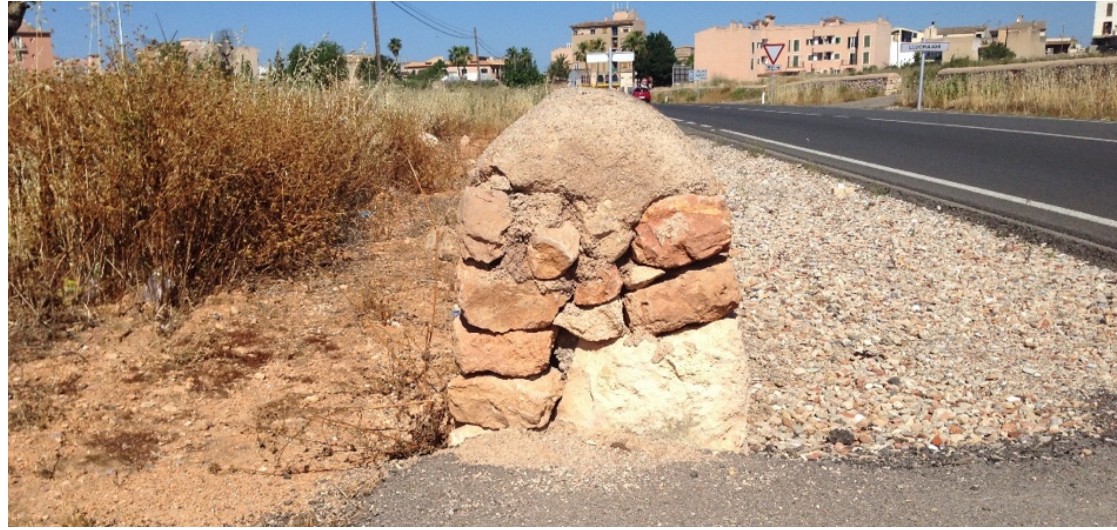

**Figure 15.** An old concave-top dry stone wall renovated with new concrete coping and chinking on the eastern outskirts of Llucmajor along route MA-19 (Figure 1). Imagery data: W. Doolittle, 2015.

Building walls with concave tops is an anomalous practice according to the construction standards of dry stone masonry in most locales [31,32]. It is not practiced anywhere else in the Mediterranean settings [33], nor is it widely recognized on Mallorca, where interest in stone work emphasizes terrace risers more than free-standing walls [34]. There exist only two explicit, but not detailed, references to water being conveyed through elevated canals or atop walls in the large body of literature on Mallorcan irrigation [18,27]. Walls that once conveyed water along their tops are yet another element of material culture on the island's cultural and hydraulic landscape that seems to have been hidden in plain sight.

### *3.4. Ingenuity and Complexity*

Of the numerous water control and other features examined on Mallorca, evidence that some, perhaps many, rock walls served as folk aqueduct bridges in times past is most compelling at a group of features 3 km southwest of the center of Campos, along route MA-6030 on the way to La Ràpita (Figure 1). West of the road by 150 m, at 39°24′42″N 2°59′31″E, are the remains of a derelict irrigation system that is more complex than the one near Algaida (Figure 16). This system involved one source that provided irrigation water that flowed over two intersecting small aqueduct bridges. No trace of a windmill, a storage tank, or even a well could be found. If a windmill existed, it might have been dismantled and the rocks pirated for construction of new buildings elsewhere, and/or perhaps used to fill in the abandoned well. It is also possible that a Persian Wheel (*noria*) might have been used to lift ground water [5]. This is evidenced by a *noria* reconstructed in 2006 that exists 2 km further down route MA-6030 at the intersection with route MA-6014, the road from Banyos de Sant Joan to Llucmajor, 39°23′38″N 2°58′54″E. Additionally reconstructed in conjunction with this *noria* is a folk aqueduct bridge that consists, in part, of ashlar tablets and, in part, of a *mamposteria* wall (Figure 17).

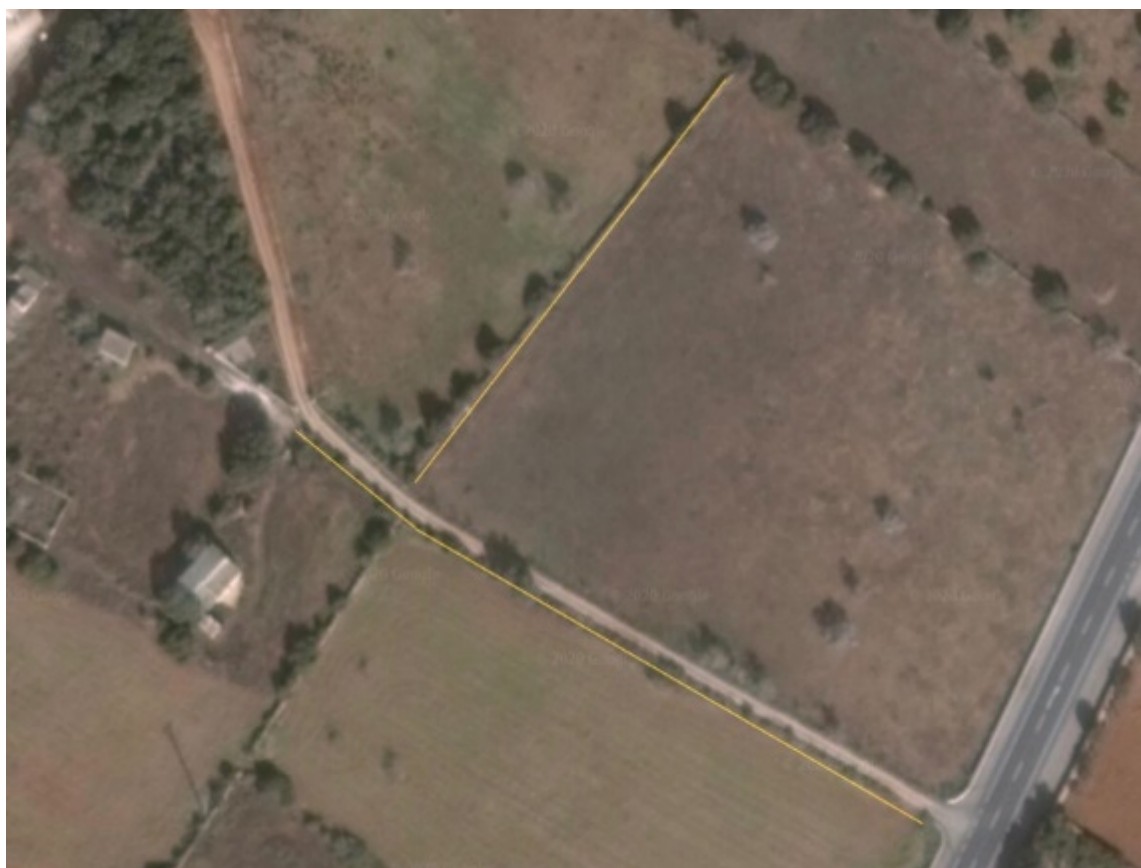

**Figure 16.** A Google Earth view of folk aqueduct bridges (yellow) south of Campos (Figure 1). Imagery data: Google, Maxar Technologies, 2020.

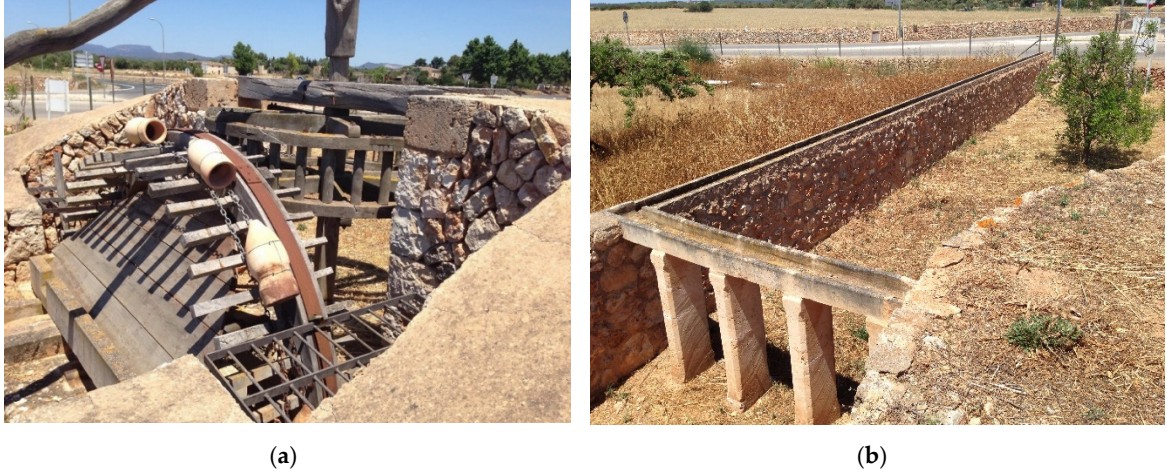

(**a**)                                             (**b**)

**Figure 17.** Reconstructed water-lifting and transportation devices 5 km south of Campos: (**a**) Persian wheel or *noria*; (**b**) two-part folk aqueduct bridge. Imagery data: W. Doolittle, 2015.

Regardless of how ground water was pumped to the surface, it flowed first atop a low *mampostería* wall. This wall comprised the last 30 m of the northeast side of an enclosure that measured ~120 m². The entire southeast, southwest, and northwest sides of the enclosure remain intact. The other 90 m of the northeast side either never existed or were removed long ago. The canal running atop the remaining wall segment/aqueduct bridge consists of imbricated ceramic tiles of the type normally used as roofing (*tejas*) (Figure 18). An interesting feature exists 20 m farther along the wall. It is a combination drop structure, flow diversion structure (*trestallador*), and header tank of an inverted siphon [35,36] (Figure 19). Other than the header tank, the inverted siphon consists of a 12 cm diameter ceramic pipe that runs north under the road and terminates at the receiving tank, which is also the beginning of the aqueduct bridge that continues north for 115 m.

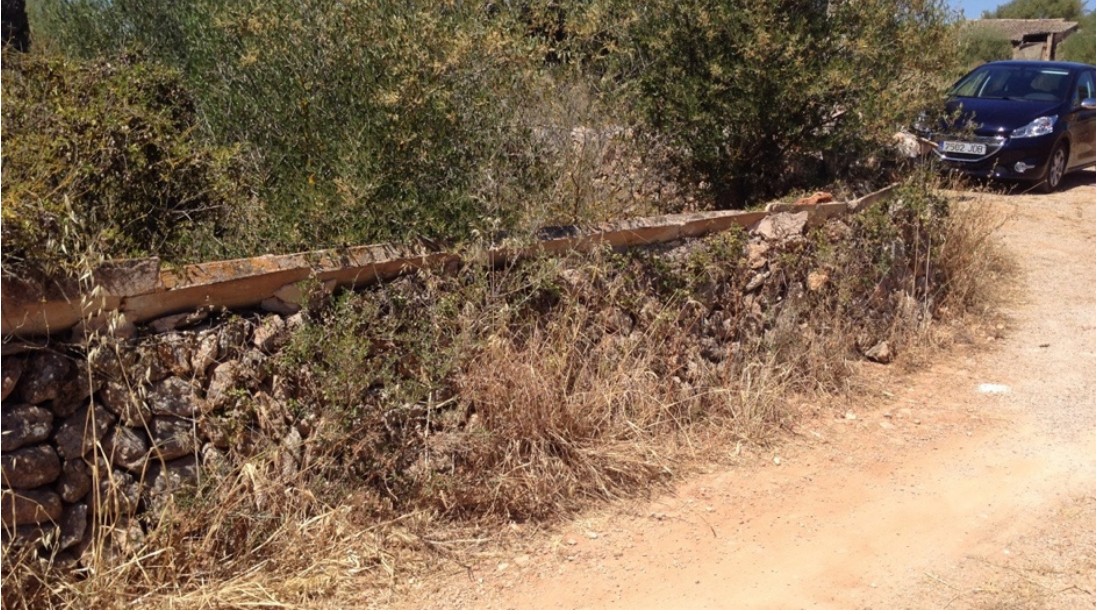

**Figure 18.** The first section of the folk aqueduct bridge south of Campos. Imagery data: W. Doolittle, 2015.

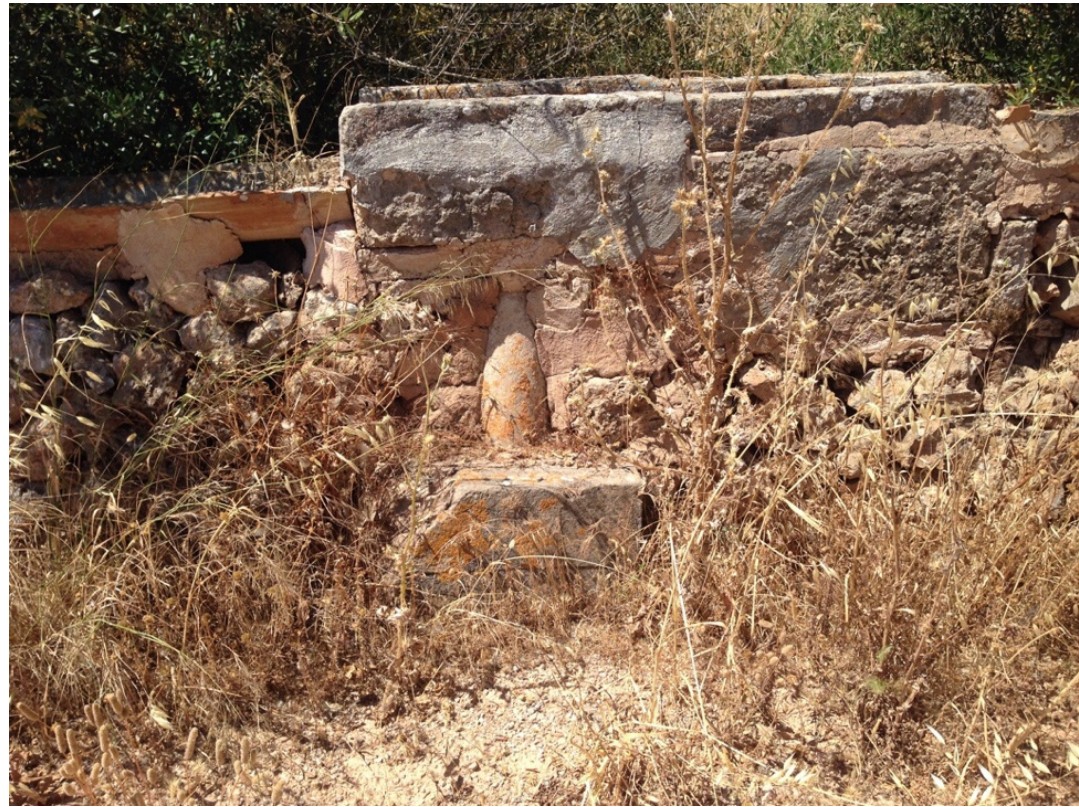

**Figure 19.** A view of the combination drop structure/flow division structure/inverted siphon header tank. Direction of flow is from right to left, and down the exposed pipe. Imagery data: W. Doolittle, 2015.

Curiously, irrigation water here did not flow atop a *mampostería* wall, but rather over an aqueduct bridge that parallels and is adjacent to such a wall. This structure is composed of crudely constructed *mampostería* piers and is topped by a canal made of modern precast concrete half pipes (Figure 20). Exactly why piers were used instead of ashlar tablets remains unknown, as does why concrete half pipes were used in lieu of ceramic roofing tiles. The differences in materials, however, illustrate the individuality of each folk aqueduct bridge, a trait that continues with the other part of this irrigation system.

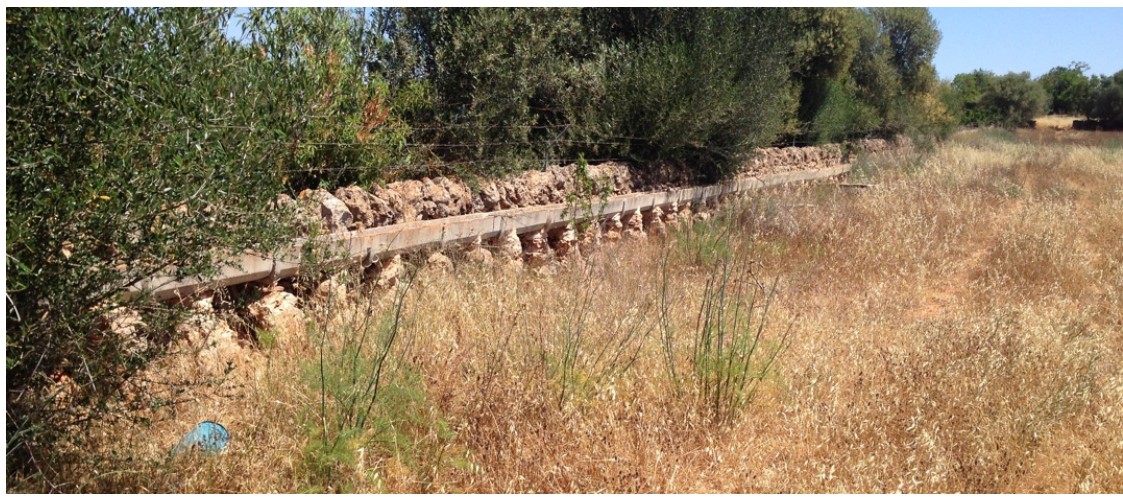

**Figure 20.** A view of the branch aqueduct bridge built of *mampostería* piers. Imagery data: W. Doolittle, 2015.

Farther on, ~10 m from the three-function feature previously discussed, and at the point where the enclosure wall turns right, the aqueduct bridge continues straight and the canal drops in height from 80 to 60 cm. In so doing, it continues not on a *mampostería* wall, but on a series of crudely cast concrete piers that support a canal of ceramic half-pipes for a distance of ~80 m (Figure 21). The concrete piers are a relatively recent renovation. The last 55 m of this irrigation system is composed entirely of traditional ceramic tiles mortared together at ground level.

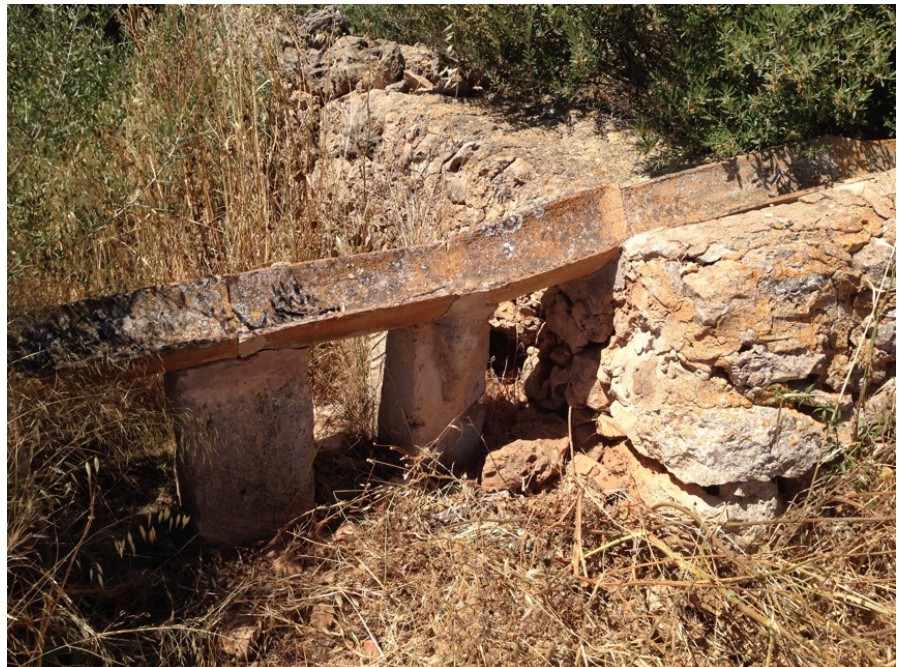

**Figure 21.** The point at which the folk aqueduct bridge changes from a *mampostería* wall to a series of recently cast concrete piers. Imagery data: W. Doolittle, 2015.

This irrigation system and all of its component parts can be clearly viewed close up in Google Earth. Using the "Street View" function, one can traverse the unpaved road from where it intersects route MA-6030 to the beginning of the folk aqueduct bridge (Figure 22).

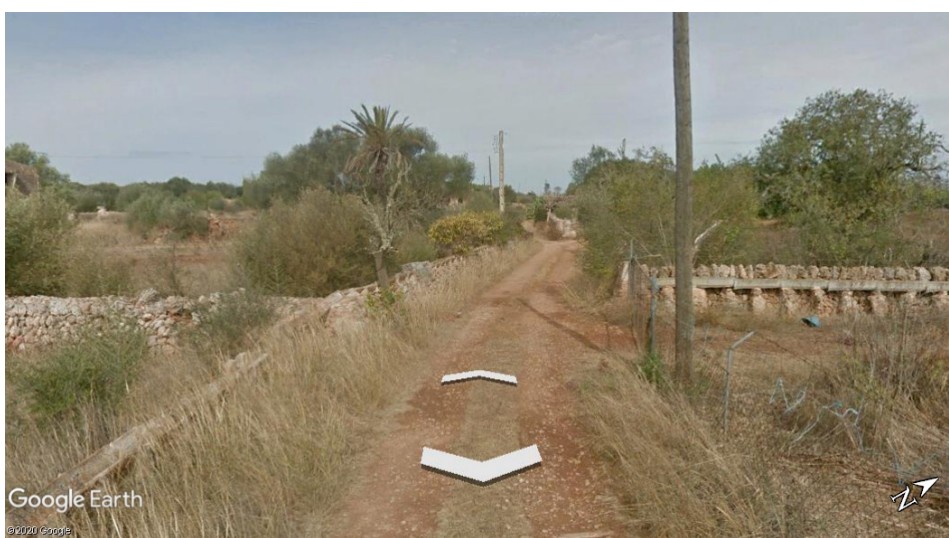

**Figure 22.** Google Earth street view image of the two connected folk aqueduct bridges south of Campos. Imagery data: Google, 2020.

### 3.5. From Mallorca to Monterey

Having existed for a very long period of time—perhaps hundreds of years—the folk aqueduct bridges of Mallorca were clearly an integral part of a sustainable agriculture system, as well as part of the island's cultural patrimony or heritage. However, there exists an additional aspect of their existence that merits consideration. As outlined in Section 1. Introduction and Section 2. Materials and Methods, a major reason for undertaking this study of water control features on Mallorca was to explore the possibility that folk aqueduct bridges in Spanish colonial California might have their origins there. Stated as a question, could the missionaries from Mallorca have carried their knowledge of transporting water atop rock walls and the building of folk aqueduct bridges to Mediterranean southern California? Or, were the folk aqueduct bridges of Mallorca prototypes for similar structures built more than 200 years ago in what is today the far southwestern United States? These questions warrant answers.

In 1769, Junípero Serra began establishing Catholic missions in California. He was born into a farm family in the town of Petra, Mallorca in 1713, ordained in Palma de Mallorca (Figure 1) and arrived in México to begin his life as a Franciscan missionary in 1749. He died in 1784 and is buried at Mission San Carlos Borromeo de Carmelo, near the city of Monterey [12].

One of the missions that Serra established is located in the present-day city of Ventura. It is Mission San Buenaventura, founded in 1782. As was the case with most of the 21 California missions, this one had an irrigation system that carried water to orchards, gardens, and fields [37,38]. The main canal for this system was 11 km long and carried water diverted by a dam from the Ventura River. The single most interesting feature of this system is the aqueduct bridge that carried water over Cañada Larga. It is located just southeast of the intersection of route CA 33 and Cañada Larga road at 34°20′31″ N 119°17′27″ W.

Being described as built of "cobbles and mortar," stone and cement," and "lime and stone," this is clearly a *mampostería* structure [39] (pp. 15–16). As is evident in Figure 23, these are entirely unshaped river cobbles. The exact length of this aqueduct bridge is unknown, as it has been heavily damaged by numerous floods [40]. Around 1900, a 4 m gap was cut through it for a new road. What remains today are two segments, one ~6 m long, the other ~21 m in length. This "stone and mortar (random rubble) wall" is supported on one side by two massive buttresses, and on the other by one. Each of these is 2 m wide and extends outward between 1.4 and 2 m [41]. These buttresses were not part of the original aqueduct bridge, but were added later to support the structure. Given that two buttresses are on the upstream side of Cañada Larga, they did not brace this structure from the forces of flood waters. That they are close to the gap between segments suggests that they were added after the road was cut through. The conclusion that the buttresses are late additions is based entirely on the obvious seams between them and the aqueduct bridge wall. Had the buttresses been part of the initial construction, there would be an interlacing of rocks between the wall and the buttresses, and no seam would be evident. The rectangular canal along the top is 76 cm wide and 25 cm deep. It was completely covered with stone slabs. At its highest point near the center of the canyon it crosses, the aqueduct bridge is 3 m high. At the downstream end of the remnant, it is 60 cm high. It doubtless was near ground level at its terminus [41].

One writer has argued that this aqueduct bridge was built between 1792 and 1815 on the basis that the governor in 1790 requested 51 artisans be sent to California from México, and among those who came were stonecutters and stone masons [42]. Two problems exist with this finding. The first problem is that skilled stonecutters and stone masons are not required to build 'random rubble' or *mampostería* structures. Mallorcans, however, had a long tradition of being skilled in this construction technique. The second problem is the failure to recognize that another Mallorcan missionary, Francisco Dumetz from Palma (Figure 1), served at Mission San Buenaventura for 15 years beginning in 1782 [12]. Long overlooked, he undoubtedly played a major role in the construction of this aqueduct bridge.

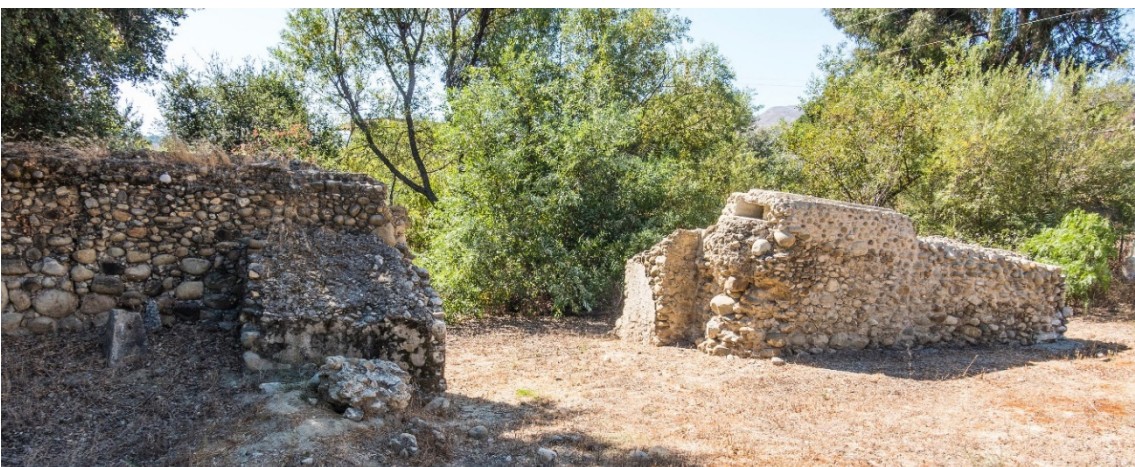

**Figure 23.** The Cañada Larga aqueduct bridge. Imagery data: Alamy 2A4FK5R.

Approximately 45 km north of Mission San Buenaventura is Mission Santa Barbara. This mission also had a water supply system that involved a dam, canals, and reservoirs, all for delivering water to the mission's gardens, fields, laundry (*lavanderia*), and fountain [43,44]. In places, the canal was elevated on arched aqueduct bridges in order to span steep-sided canyons. In a few places, however, the canal was laid atop *mampostería* walls. Josephine Clifford noted in 1872 that "a wall, from eight to fourteen feet high, was built of rock and cement, and on the wall, encased in hard cement, lay pipes, burnt of clay, after the manner of tiles, each pipe about ten inches long, narrow at one end and wide at the other, to insert the next piece" [43] (p. 1). In addition to clay pipes, roof and paving tiles were also used [45].

Remnants of a canal "atop stone walls to maintain appropriate gradient" and an "elevated aqueduct" can be found 235 m north of the Santa Barbara Mission church along Mountain Drive above the intersection with E. Los Olivos Street, 34°26′25″ N 119°42′44″ W (Figure 24) [43]. However, closer to the church, a mere 42 m northeast of its main door, on the other side of E. Los Olivos Street, is a much larger segment of a raised aqueduct bridge that once conveyed water to the mission's orchard, tanning vat, and laundry, 34°26′19″ N 119°42′45″ W (Figure 25) [43]. This structure, and the smaller segment upstream, were made of sandstone river cobbles, probably gathered from the nearby Mission Creek. Large rocks were used near the base, and smaller rocks were used near the top. Chinking included smaller stones and chunks of fired clay. Mortar was doubtless used during construction, and concrete was added relatively recent to stabilize the structures (Figure 26). It is 1.4 m high where it is was cut for the street. At its end, 91 m away, it is ~3 m high. It varies in width between 69 and 84 cm, and the depth of the channel varies from 30 cm at its head and 23 cm at its end [43]. The entire water control system was planned in 1799 and completed by 1808 [45]. The missionary who is credited with building this system was not from Mallorca. However, Francisco Dumetz, the Mallorcan from Mission San Buenaventura, visited Mission Santa Barbara for a month in 1792 [46]. While there, he may well have planted the idea of the aqueduct bridges to the then-resident missionary who surely knew of the one at Mission San Buenaventura.

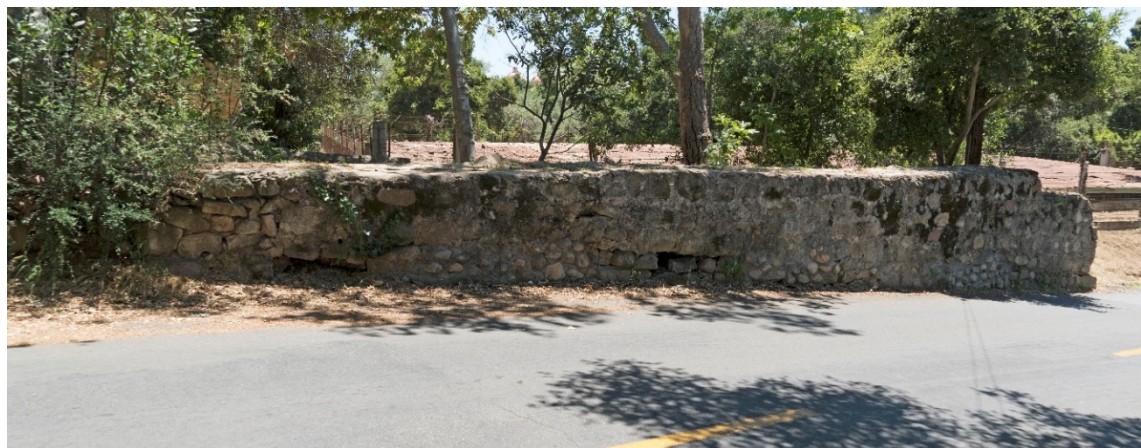

**Figure 24.** A remnant of a wall on top of which once flowed irrigation water. Imagery data: M. Glassow, 2020.

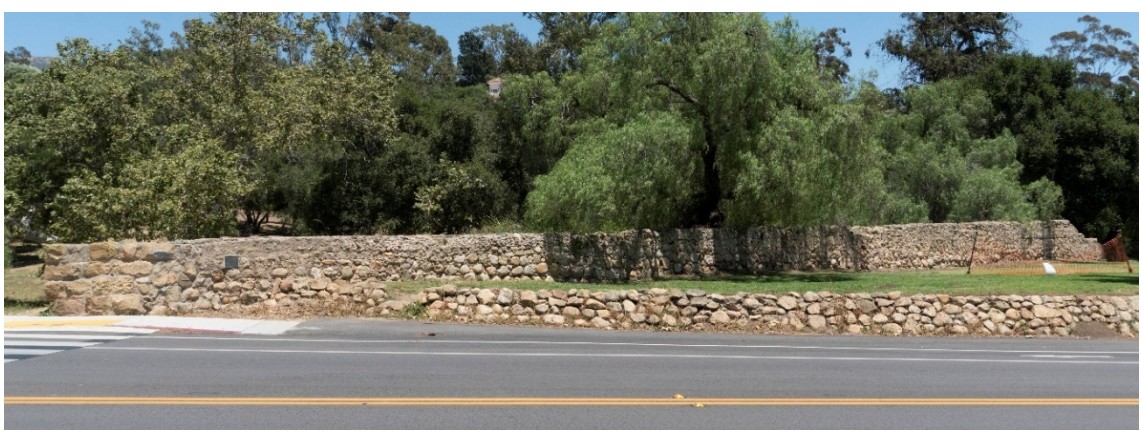

**Figure 25.** The aqueduct bridge at Mission Santa Barbara. Imagery data: M. Glassow, 2020.

Mallorcans were major players in the California mission system. Of the 142 missionaries who served there, 16, or 11.3%, came from Mallorca [12] (Figure 1). Of Spain's 50 provinces, only Cantabria and Catalonia provided California with more missionaries than Mallorca. The influence of Mallorcans in constructing irrigation systems and related features pervades the California missions [47]. In addition to the folk aqueduct bridges at San Buenaventura and Santa Barbara, a few other cases stand in evidence. Two Mallorcans co-founded Mission San Antonio de Padua in 1771. These were Miguel Pieras from Palma and Buenaventura Sitjar from Porreras [12] (Figure 1). Together, they initiated construction on the first irrigation system there, which was the first irrigation system in California [48]. Both of these missionaries were replaced in 1804 by yet two more Mallorcans, Pedro Cabot from Buñola and Juan Bautista Sancho from Artá [12] (Figure 1). Together, they expanded, improved, and completed the irrigation system [48,49]. Finally, Mariano Payeras from Inca, Mallorca is credited with having completed the irrigation system at Mission Purísima Concepción sometime between 1804 and 1811 [12].

The construction of irrigation systems, including bridge aqueducts, has long been regarded as an important aspect of California mission history. Similarly, the role of Mallorcans has long been appreciated by California historians. The connection between the two, however, has escaped attention. For example, in speaking of water control missionaries at Mission San Antonio de Padua, one writer noted: "Much is to be attributed to padres at the mission who laboriously worked to provide water for it" [38]. He clearly did not recognize that these missionaries were from Mallorca. Until now, no one could have known that the Mallorcans brought with them, from their homeland, the practice of conveying water atop rock walls and the construction of walls for the purpose of transporting water.

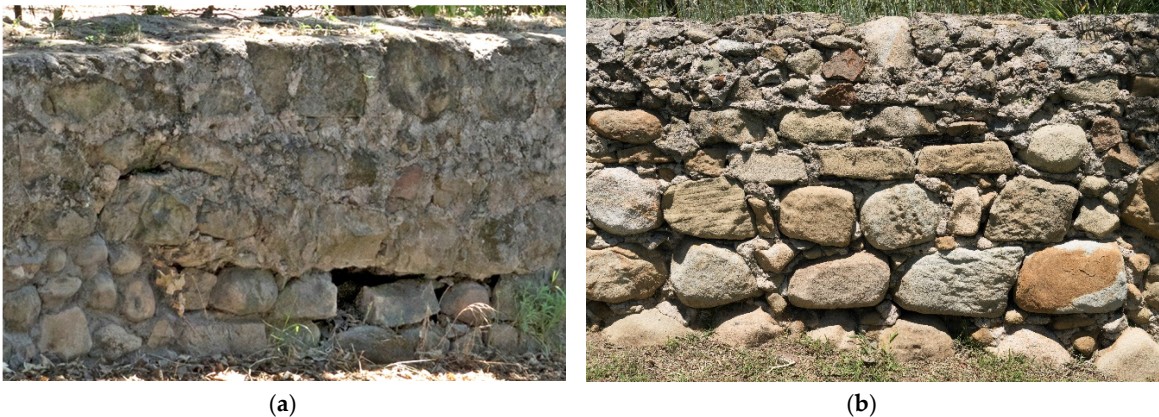

(**a**)　　　　　　　　　　　　　　　　　　　　　　(**b**)

**Figure 26.** Close-up views of two segments of the Santa Barbara folk aqueduct bridge: (**a**) The upstream segment (Figure 24); (**b**) the downstream segment (Figure 25). Compare with Figures 14, 18 and 21.

## 4. Discussion

Water-lifting mills and irrigation on Mallorca are mentioned in the written records dating to the Catalonian conquest of 1287 [35], with very little change having occurred in local agriculture until the arrival of tourists in 1960 [27]. Drip irrigation systems (*rec per degotei*) in orchards (*huertas*) and the use of soaker tubes (*tub exudant*) in fields then became popular among farmers on the dry eastern plains [27]. Small aqueduct bridges began to fall out of use, and dry stone walls separating fields became places to store irrigation pipes (Figures 13b and 14). The demise of folk aqueduct bridges is no more evident than in the Figures 11, 12 and 18, Figures 19–22, where grasses have grown up around the structures. If these irrigation systems were in use or used recently, there would be no such vegetative growth, as farmers would have eliminated weeds that consume precious water and undermine the integrity of the structures.

Whereas all but one windmill on the island was in poor shape and inoperable in 1997, in 2015, perhaps as many as 20% have been renovated, and more are in the process of renovation. According to an informant in the rural community of El Palmer, windmills are being renovated with the aid of government subsidies, all for the purpose of generating electricity rather than lifting water. Along route MA-6040 from Campos to Colònia de Sant Jordi (Figure 1), one is never out of sight of a renovated electricity-generating windmill.

Windmills might be making a comeback, but it is doubtful that many folk aqueduct bridges will be renovated. There is no longer a need to convey water through canals, and indeed, the cultivation of grains may well stop in the not-too-distant future. If these structures are to be saved as part of the island's unappreciated heritage, it could be in the form of landscape ornamentation on the increasing number of farm stays (*agroturismos*). Old farmsteads are rapidly being converted into boutique hotels, resorts, and spas, all of which attempt to capture the ambience of farm life in times gone by [50].

If there is a single characteristic of these folk aqueduct bridges that gives them "character," it is that they are all different. They all share a basic design with a gradient predicated on gravity flow [7]. However, in terms of structural materials: Some have *mampostería* walls; some have ashlar tablets; some employ cast-in-place concrete piers; some have *mampostería* piers; and some are comprised of combinations of each. In terms of superstructural materials: Some folk aqueduct bridges have rectangular tablet canals; some have imbricated roofing tiles; some have ceramic half-pipes; some have precast concrete half-pipes; and some are comprised of combinations of each. These aqueduct bridges are indeed "folk" structures [4] (pp.33–38).

Folk, or common people, have long been recognized as agents of technology transfer from one continent to another [51]. Mallorcan missionaries in Spanish California are among such folk. Their previously unrecognized roles in transferring water control technology are now clear. The California missions are today seen as part of the state's and the country's heritage. These missions

functioned for more than a century, feeding thousands of people. They were clearly sustainable, as were their predecessors on Mallorca.

## 5. Conclusions

Along with windmills, rock walls dominate the cultural landscape of Mallorca to the extent that nuanced details of their functions, and less conspicuous but related features, have been long underappreciated. This paper identifies and describes these features and discusses their significance. In doing so, it has five major findings. The first is that folk aqueduct bridges made of locally available materials have existed for a very long time, and contributed to the sustainability of agriculture on Mallorca. The second finding is that they are falling out of use and, as such, deserve historic preservation because of their cultural patrimony. The third finding is that many rock walls that appear today as boundary markers and/or field enclosures also served in times past to transport water along their tops. This water was lifted from wells by means of windmills or Persian wheels, and delivered to fields of wheat and perhaps olives. In effect, some walls were themselves folk aqueduct bridges. The fourth finding is that the folk aqueduct bridges of Mallorca were prototypes of those constructed in the late 18th century Spanish colonial California by missionaries from the island.

In addition to these four factual findings, this paper has one final conclusion of a broad intellectual nature. That is, one should never accept what meets the eye at first glance. Many features and their implications are hidden in plain sight. Looking is not always seeing. Seemingly mundane features can possess qualities that need to be exposed and illuminated.

**Funding:** This research was funded by a grant from the Houston Endowment through the Institute of Latin American Studies, The University of Texas at Austin, and with funds from the Erich W. Zimmermann Regents Professorship, The University of Texas at Austin.

**Acknowledgments:** This paper is dedicated in honor of Elisabeth K. Butzer, to the memory of Karl W. Butzer, and Terry G. Jordan-Bychkov, three of the best friends and colleagues a scholar could ever hope for. I thank Helena Kirchner for her insights and information on studies of medieval water control on Mallorca; James G. Mills, Kim Hocking, Michael H. Imwalle, and Robert L. Hoover for their insights and information on water control at the Spanish missions of California. Special thanks go to Francisco Ochoa for crafting Figure 1, and Michael Glassow for the images that appear as Figure 24, Figure 25, Figure 26. Rights to reproduce all images have been obtained.

**Conflicts of Interest:** The author declares no conflicts of interest.

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
