# Peer review of "Stacking Rocks to Transport Water: Folk Aqueduct Bridges of Mallorca and Spanish Colonial California"

_sustainability, doi:10.3390/su12135257_

Round 1

Reviewer 1 Report

Thank you for asking me to review this fascinating paper.

The material is not only relevant to the research questions about Mediterranean landscape and Transatlantic linkages addressed by the author but will serve as a resource for future scholars as the Mallorca landscape inevitably changes.

One additional point to make is that most of the colonizers of New Spain were Andalusian, with Seville and Cadiz serving as major connecting nodes into the Atlantic world. Catalan ports were shut out of the Atlantic world until the nineteenth century. That makes this particular influence on Californian landscape quite unusual and shows how important the clergy were in forming the landscapes of New Spain. Clergy were more diverse than the general colonizers of New Spain and, as this research confirms, played influential roles.

Another point to explore is that the Romans and Arabs occupied Mallorca for centuries before the Catalans. What was their influence on these aqueducts? I know that is not the focus of this research, but it's a question a lot of readers will be curious about. A sentence or two that addresses that connection to the deeper past would be interesting, I think, to many readers.

A few presentation suggestions:

1) The extensive use of Google imagery of several types is nice but needs to be attributed to avoid copyright infringement. Google is quite specific on the requirements. See https://www.google.com/permissions/geoguidelines/attr-guide/

2) An original map would be a welcome addition. Many readers who will be interested in this article will be unfamiliar with the geography of Mallorca and its location in the Mediterranean. Some will be interested in revisiting the same study sites for future research, so the maps could show them as well, besides the already provided latitude, longitude coordinates.

3) There are some spelling typos that need to be corrected.

Author Response

Thank you for these excellent and very helpful comments.

  1. I will add a few sentences on the Spaniards going to the New World, a la the work of Peter Boyd-Bowman, and the emphasis on missionary influences.
  2. Similarly, a note about Roman and Arab influences on Mallorca will be added.
  3. The Google Earth images will be replaced as soon as Covid-19 allows me to fly to California. The pictures I took earlier are of low quality, and I included GE images as a temporary means of illustrating my point.
  4. A map will be added.
  5. And, typos will be fixed.

Again, thank you.

Reviewer 2 Report

Assessment:

This is a low-quality article about the sustainability of a folk heritage in Mallorca: the small aqueduct bridges built by individual farmers, frequently unappreciated. This heritage and landscape in Mallorca could be interesting. However, author give more a personal “panegiricus” around Mallorca, that a real analysis and evaluation of this kind of heritage. Lack the individualisation of an objective and an appropriate research design.

Recommendations.

A.- STRUCTURE.

Article structure could be improved. I recommend follow Sustainability template.

1.- Introduction.

Introduction doesn't give the goal and point of depart of the research.

2.- Materials and Methods.

Materials and methods chapter must be developed. This chapter it´s a litter summary of the author´s activities in Mallorca, but not a real exposition of the development of the research. Lack the exposition of the systematic survey or investigation of aqueduct bridges etc cited. It isn´t interesting when author arrive in Mallorca and some similar subjective or personal comments.

3.- Results

3.1. The Mallorcan Landscape

Lack geographical sources of this environment. I recommend use the geological maps and similar data of maps IGME-Spain.

3.2. Building Materials and Techniques

It could be interesting cite the chronological date, approximate, of several examples of ashlar walls presented. Figure 5, yes, is an old wall made of hand-sawed ashlar tablets. These ashlar tablets seem reemployed in a wall. Its original provenance, could be roman? Medieval?  After? 

Article need a more accurate explanation of the history of construction techniques. This point could be developed in materials and methods chapter.

3.3. The Initial Discovery

126-127: “Figure 6, is located 2 km northeast 127 of the center of Algaida, off of route MA-3131 on the way to Sant Joan”. Please, place all of your findings in a map. Use Google Earth. Or, better, use the IGN-Spain webpage. You have all historical orthophotos since 1945. Maps since 1895.

4.- Discussion / Conclusion

There are no conclusions. Only a general repetition of the comments gave in introduction chapter. There is a little cite to missionaries of Spanish California without any connection with the argument of the main paper, the aqueducts in Mallorca. Title says: Bridges of Mallorca and Spanish Colonial California. However, the second it isn´t developed.

B.- IMAGES

Lack an appropriate graphic documentation. Lack drawings of the different construction techniques, comparison between them, location on the maps.

Author uses Google Street View too much for his analysis. This source doesn´t allow appreciate the techniques and other details. It could be desirable a personal documentation.

Author Response

Thank you for your comments on my ms. Most will be addressed and evident in the next draft. A few, however, may not be possible.

  1. I agree the intro can be improved by adding the goal of this research.
  2. Similarly, I will expand on how this research was developed. My survey was systematic, albeit not in the usual sense. Once I made my initial discovery, I literally drove every road on the eastern plains looking at walls, windmills, and related features.
  3. Maps will be added as per 3.1 and 3.3.
  4. The chronology of walls and history of construction techniques is beyond the scope of this study. I do, however, cite the relevant literature, such as it is, on these topics.
  5. I was stunned by the comment "There are no conclusions" until I re-read my paper and saw that there weren't. I'll fix that for sure!
  6. Google Earth images were used only because I can't fly to California due to Covid-19. Pictures I took earlier during my field work are of poor quality. New pictures will replace the temporary GE images.

Reviewer 3 Report

The manuscript is interesting but has more of a relationship story than a scientific article. One should consider whether it is a scientific article or, for example, a technical note or other type of scientific writing.

Introduction should end with a paragraph with the main aims of the article.

Line 61-63: “I was struck immediately by a rural landscape characterized by "a forest of windmills" [12] (p. 281) that once pumped irrigation water to otherwise dry fields” - The first impression would be more prominent by using photos. Please add some photos if the author has them.

In the text of the article, there are few statements and comparisons of objects in Majorca and California built by the same Spanish community. It is recommended to add some combinations, or as the Author will - pictures from paralytic objects in California.

Delete references in the manuscript to pages in the bibliography (e.g. Line 62, Line 81) etc.

Author Response

Thank you for your brief but spot-on comments.

  1. The paper is more of a relationship story than a scientific article. I will endeavor, however, to frame it more in a science format. That said, however, there are human, cultural, and historical dimensions to sustainability that this paper attempts to portray.
  2. The intro definitely needs and ending that points out the article's aim. It will be added.
  3. Try as I might, being a not-so-skilled photographer, to capture the "forest of windmills" on film, I failed. My pictures simply didn't do the trick. But, you've inspired me to search for some online. If I find one, I'll purchase the rights to use it, and I'll include it.
  4. I intend to make another trip to California as soon as this Covid madness ends. My goals is to take some better pictures of features discussed in the paper. I relied on a few Google Earth images as a temporary means of illustrating features. They will be replaced.
  5. Reference will be corrected.

Round 2

Reviewer 2 Report

Dear Author, you have worked very well. Thank you very much for your time and effort. 

  • I still recommend you to use the information from the IGME (Geological National Institut of Spain-http://www.igme.es/) for geological characterization. Documentacion is completly freely for download citing the copyright. 
  • You must review the layout of images. Some images are displaced from the caption. Letters have also been moved. As an example Fig. 8, Fig. 17.
    - I remember that you must to have the permissions to reproduce the images of other authors or services. Do you have permission for figure 2?
  • - Line 330: Is Monterrey, not Monterey (double rr)

Author Response

As per comments of Reviewer 2:

Instead of the IGME characterization, I prefer the more simplified and generalized geological classification that I borrowed from the articles I cite in the text. I guess we disagree, but if it was acceptable for the articles, and hence geological and geographical journals, that I cite, it should be acceptable here. Were this article specifically about geology, I might well have used the more specific IGME classification.

The displacement of captions to some figures, e.g., 8 and 18 are merely a function of page spacing. As I noted in my letter, I am leaving it up to you or your editorial staff to reposition the images as needed in the final format. If, however, you want me to get them on one page, I will do so.

As stated in the Acknowledgments, I have permission to publish all images.

Monterey, California is spelled with one "r," unlike Monterrey, Nuevo Leon, Mexico.

W. Doolittle

Reviewer 3 Report

The manuscript has been significantly improved and now warrants publication in Sustainability.

Author Response

Reviewer 3.

  1. You are correct, the paper was more of a relationship story than a scientific article, and on reflection not an appropriate one. I corrected this...to your satisfaction, I hope. To the Introduction I have added the goals of this research. See lines 57-65 and 71-74. I have also totally rewritten the Materials and Methods section, outlining in detail how I conducted surveys and carried out detailed studies of various features.
  2. I found and added a "forest of windmills" image.
  3. I have added new images of features in California that are appropriate for comparisons to features on Mallorca, particularly the new Figure 27.
  4. I did not delete all references to pages in the text. The Instructions for Authors details how to reference page numbers, and I followed those instructions to the letter. I used page numbers only in those cases where I cite a direct quote.